# Humoral Immune Response of Mice against a Vaccine Candidate Composed of a Chimera of gB of Bovine Alphaherpesviruses 1 and 5

**DOI:** 10.3390/vaccines11071173

**Published:** 2023-06-29

**Authors:** Juan Sebastian Quintero Barbosa, Carlos Javier Alméciga-Díaz, Sandra E. Pérez, María Fernanda Gutierrez

**Affiliations:** 1Virology Laboratory, Infectious Diseases Group, Microbiology Department, Faculty of Science, Pontificia Universidad Javeriana, Bogotá D.C. 110231, Colombia; 2Institute for the Study of Inborn Errors of Metabolism, Faculty of Science, Pontificia Universidad Javeriana, Bogotá D.C. 110231, Colombia; 3Tandil Veterinary Research Center (CIVETAN)-CONICET, Faculty of Veterinary Sciences, National University of the Center of the Province of Buenos Aires, Tandil B7000GHG, Argentina

**Keywords:** candidate vaccine, *Bovine alphaherpesvirus*, glycoprotein B, neutralizing, antibodies, Bovine Alphaherpesvirus 1 and 5

## Abstract

Infectious bovine rhinotracheitis (IBR) and bovine meningoencephalitis are caused by *Bovine alphaherpesvirus* (BoHV) types 1 and 5, which seriously threaten the global cattle industry. Vaccination to improve immunity is the most direct and effective means to prevent these conditions. Glycoprotein B (gB) is essential for the attachment of both viruses to permissive cells, and is a major target of the host immune system, inducing a strong humoral response. The aim of this study was to evaluate, in a murine model, the immune response of a candidate vaccine formulation composed of a chimeric BoHV-1 and BoHV-5 gB (DgB), expressed in *Komagataella phaffii*. The chimeric DgB vaccine adjuvanted with Montanide 50 ISA V2 or aluminum hydroxide was administered intramuscularly or subcutaneously. A control group and a group that received a commercial vaccine were inoculated subcutaneously. Higher titers of neutralizing antibodies against BoHV-1, BoHV-5, and a natural BoHV-1/5 recombinant strain were obtained with the oil-based candidate vaccine formulation administered intramuscularly. The results demonstrated that the chimeric DgB conserved important epitopes that were able to stimulate a humoral immune response capable of neutralizing BoHV-1, BoHV-5, and the recombinant strain, suggesting that the vaccine antigen is a promising candidate to be further evaluated in cattle.

## 1. Introduction

*Bovine alphaherpesvirus* (*BoHV*) 1 and 5 belong to the family *Herpesviridae*, subfamily *Alphaherpesvirinae* and genus *Varicellovirus* [1]. BoHV-1 is the causative agent of genital infection, abortions, encephalitis, and infectious bovine rhinotracheitis (IBR) [2], whereas BoHV-5 is mainly responsible for meningoencephalitis in calves [3]. After the primary infection by these viruses is solved and non-clinical signs are seen in the animals, the infection persists in the host throughout their life in a state of latency in the trigeminal or sacral nervous ganglia, causing permanent infections in cattle herds [4]. Sporadically, the latent virus can be reactivated and transmitted to other susceptible hosts [5].

*Bovine alphaherpesviruses* 1 (BoHV-1) and 5 (BoHV-5) are important pathogens around the world [2]. Both belong to the Herpesviridae family of viruses, the alphaherpesrvirus subfamily and within the Varicellovirus genus, and are closely related since BoHV-5 was initially reported in Australia as a neuropathogenic variant of bovine herpesvirus type 1 [1]. Several studies of these infectious agents, which included restriction site mapping of viral DNA, monoclonal antibody reactivity, and cross-neutralization tests, showed that these two virus strains differ in both their genetic and serological properties. Thus, the ICTV in 1992 reclassified it as BoHV-5 [1].

These two species present subspecies; in the case of BoHV-1 there are three viral subtypes, BoHV-1.1, BoHV-1. 2a, and BoHV-1. 2b, that were characterized by restriction endonuclease patterns, while three subtypes of BoHV-5 have also been found, BoHV-5a, BoHV-5b, and BoHV-5non-a/b [1,2,3].

Although BoHV-1 and BoHV-5 are genetically and antigenetically related, they have important differences in their ability to neuro-invade. This ability of BoHV-1 generally does not progress beyond first-order neurons (trigeminal ganglion), where it remains in latent infection, unlike BoHV-5, which manages to ascend through the central neuro system and infect brain regions [4,5].

The structural components of BoHV-1/5 include a double-stranded DNA genome enclosed in an icosahedral nucleocapsid surrounded by the tegument, an amorphous layer with some structured regions. Finally, BoHV virions are covered by a lipoprotein envelope that contains 10 glycoproteins important for the viral life cycle [1].

Currently, there are several vaccine alternatives, including traditional (inactivated and attenuated vaccines) and new generation vaccines (recombinant, vectored-, and DNA-vaccines) [6]. Although the World Organization for Animal Health (OIE) reports that vaccination against BoHV-1/5 can effectively reduce the clinical manifestations and economic losses, these vaccines do not completely protect against infection, latency, and virus reactivation [6]. Therefore, new vaccine technologies against these alphaherpesviruses are required.

Recent reports on the development of vaccine candidates have focused on viral glycoproteins due to their location (i.e., envelope) and importance in the viral life cycle [7,8,9,10,11]. In particular, BoHV-1 and BoHV-5 have the same number of glycoproteins that share a high degree of structural and functional homology [12,13]. Herpesviruses glycoproteins are responsible for the initial adhesion of the virus to cellular receptors and the subsequent penetration into the host cells, influencing cell tropism [14,15,16]. In addition, glycoproteins participate in several processes such as capsid envelopment, viral release, and the transmission of infection by cell-to-cell dissemination [14,15,17].

BoHV-1/5 glycoprotein B (gB) plays an essential role in the first steps of the viral life cycle, such as adhesion [18,19,20], facilitating the penetration, and subsequent stages of the cycle. Previous studies have shown that the immune response is directed to this specific glycoprotein [13]. In this sense, stimulating the production of neutralizing antibodies against gB may contribute to the prevention of the subsequent life stages of the infectious cycle, such as latency and reactivation [10]. The use of glycoprotein B seeks a humoral immunity capable of reducing a primary infection from a quantitative and qualitative expression of neutralizing antibodies, preventing BoHV-1 and -5 from accessing latency sites. Looking for a product capable of increasing protection time with as little administration as possible, we performed the immune-informatic characterization of a chimeric protein composed of glycoprotein B domains from BoHV-1 and BoHV-5, obtaining high scores in the different searches for epitopes recognized by B and T lymphocytes. This vaccine candidate was produced by recombinant techniques in the yeast *Komagataella phaffii*, which was recognized by a monoclonal antibody from a commercial ELISA kit used for IBR diagnostics, suggesting that the epitopes are conserved among the entire infectious virus [21]. Due to its ease and lower cost, yeasts are the main candidates for the production of glycoprotein subunits as vaccine candidates; in addition to this, being a eukaryotic organism, post-translational rearrangements such as disulfide bridges, glycosylations, foldings, etc., are important for the antigenicity of the candidate [9,11,21].

Thus, the aim of this study was to evaluate whether the chimeric BoHV-1/5 gB vaccine candidate expressed in the yeast *K. phaffii* induces neutralizing antibodies. For this purpose, the production of neutralizing antibodies against BoHV-1, BoHV-5, and a natural recombinant strain induced by the vaccine candidate formulation containing oil or aluminum hydroxide adjuvants and a commercially available vaccine was evaluated in a murine model.

## 2. Materials and Methods

### 2.1. Expression of Chimeric BoHV-gB Candidate Vaccine

The chimera gB vaccine candidate, composed of PH-like domain 1 from BoHV-1 and PH-like domain 2 from BoHV-5 (hereafter DgB), was expressed in *Komagataella phaffii* GS115 and produced as previously described [21,22,23]. After induction phase, the recombinant protein was concentrated from the culture supernatant by using Centriprep 50YM (Millipore, Burlington, MA, USA) and purified by affinity chromatography using both HisTrap™ HP 1 mL columns pre-packed with pre-charged Ni Sepharose™ and the ÄKTAprime™ Automated Liquid Chromatography system (GE Healthcare, Chicago, IL, USA). Protein concentration was determined by Pierce BCA protein assay (Thermo Fisher Scientific, Waltham, MA USA), according to the manufacturer’s instructions. The chimera DgB was lyophilized and used for the different vaccine formulations.

### 2.2. Cells and Viruses

Madin Darby bovine kidney cells (MDBK, ATCC^®^ CCL-22™) were grown in Dulbecco Eagle’s Minimal Essential Medium (DMEM) (Biowest, France) supplemented with 10% fetal bovine serum (Biowest, Nuaillé, France) at 37 °C in a 5% CO_2_-humidified atmosphere. The infective titer of the stocks of BoHV-1 (Cooper strain), BoHV-5 (97/613), and the recombinant BoHV-5/1 virus (A663) was determined by microtitration test in MDBK cells by performing serial ten-fold dilutions. Fifty µL virus suspensions/well were inoculated in 96-well plates onto monolayers of MDBK cells. After 3 days of incubation, the viral stock titer was determined by the method of Reed and Muench (1938) and expressed as log10 TCID50/mL. BoHV stocks were stored in liquid nitrogen.

### 2.3. Animal Experiments

Six groups of six outbred Swiss albino female mice, of six-to-nine weeks of age were used for this experiment. All mice were maintained and handled at the animal care center of Universidad Nacional Del Centro De La Provincia De Buenos Aires (UNCPBA, Tandil, Argentina), and housed in autoclaved cages with no food or water restrictions. The procedures were performed in accordance with the guidelines of the Animal Welfare Committee, FCV, and UNCPBA and approved by the UNCPBA Animal Research Ethics Committee (Animal Welfare Act ResCA 087/02).

### 2.4. Immunizations

Mice in each group received three doses of the corresponding vaccine formulations, with 14-day intervals. Vaccines formulated with 50 μg DgB adjuvanted with Montanide 50 ISA V2 50% *v*/*v* (Seppic Adjuvants, France) or aluminum hydroxide (HA) were administered intramuscularly (i.m) or subcutaneously (s.c). Control groups were s.c inoculated either with PBS or with an inactivated commercial vaccine (Strain Copper for BoHV-1 and Strain 663 for BoHV-5, both strains ≥ 10^7.5^ DICT _50_) following the manufacturer’s instructions. Mice were observed daily, and blood was obtained by retro-orbital bleeding every 14 days after primary vaccination until day 42. 

### 2.5. Virus Neutralization Test (VNT)

Serum samples were analyzed for BoHV-1 neutralizing antibodies on day 0, 14, 28, and 42. On days 14 and 28, serum samples from each group were pooled due to the low quantity of total blood obtained per mouse. On the other hand, neutralizing antibodies against BoHV-5 and the recombinant strain were only analyzed on day 42. Serum was stored at –20 °C until use. The presence of neutralizing antibodies was evaluated by the serum neutralization test using the microtiter technique in 96-well plates on MDBK cells. After heat inactivation (30 min at 56 °C), serial 2-fold-serum dilutions were performed and a viral inoculum containing 100 TCID50/50 µL was added. After 30 min incubation at 37 °C, MDBK cells were added. Controls for virus TCID50/mL and BoHV positive and negative serum controls were included in each test. The antibody titer was determined as the highest dilution that showed complete inhibition of the cytopathic effect (CPE) after 72 h of incubation.

### 2.6. Statistical Analysis

Data were analyzed using Rstudio for Mac OSX 10.15 (RStudio, PBC, Boston, MA, USA). Analysis of differences in serum neutralizing antibodies among treatment groups was performed on log2-transformed titer data. Statistical differences in serum neutralization antibodies were investigated using one-way ANOVA followed by Dunnett’s multiple comparisons test. Data were compared to non-treated control and commercial vaccine groups. Significant differences were considered when *p* ≤ 0.05.

## 3. Results

### 3.1. Expression of Chimeric BoHV-gB Candidate Vaccine

Production of the recombinant vaccine candidate was performed in *K. phaffii* GS115. Secreted recombinant protein was detected by dot blotting at 72 h post-induction. The recombinant candidate vaccine (DgB) concentration obtained with this process was 873 µg/mL.

### 3.2. Mice

The animals were checked every day that the experiment lasted, and at no time did we observe adverse reactions to either intramuscular or subcutaneous inoculation. Mice showed normal behavior and had normal food and water intake.

### 3.3. Humoral Immune Response

The humoral immune response was evaluated by a virus neutralization test, as previously described. For BoHV-1, serum samples obtained on days 14, 28, and 42 following primo-vaccination were evaluated, whereas for BoHV-5 and the recombinant virus, only 42-day serum samples were evaluated since the quantity of serum obtained was not enough to perform the test with all BoHV strains. The evaluation of the serum neutralization test against BoHV-1 at all time-points was based on the higher prevalence of the virus in Colombia when compared to the other strains included in this study [6]. 

The results of the viral seroneutralization against BoHV-1 carried out on day 14 post-inoculation showed that the only formulation with a detectable antibody titer belonged to DgB + ISA 50 V2; the commercial vaccine did not induce neutralizing antibodies. The data were transformed into Log2 as can be seen in Figure 1a. At 28 days post-inoculation, neutralizing antibodies were detected in mice that received DgB + ISA 50 V2 (s.c), and DgB + aluminum hydroxide (i.m) and DgB + ISA 50 V2 (i.m) stimulated an even higher antibody titer; in the case of the commercial vaccine no neutralizing antibodies were shown, and neutralizing antibodies were not detected in the PBS-inoculated group. These data were also transformed into log2 as can be seen in Figure 1B.

In the viral neutralization test carried out on day 42 after the first immunization (Figure 2), all vaccine formulations induced neutralizing antibodies. In this case, a statistical analysis of a one-way ANOVA followed by a Dunnett’s test of multiple comparisons of the treatments against the control (PBS) were performed. Nevertheless, only the formulation with DgB + ISA 50 V2 (i.m) had significant titer (*p* < 0.01) when compared with the PBS-vaccinated control. In mice that received the commercial vaccine, neutralizing antibodies were detected, although statistical differences with the control were not detected (*p* ≥ 0.05).

The serum neutralization test against BoHV-5 was performed 14 days after the last vaccination (Figure 3). In this case, a statistical analysis of a one-way ANOVA followed by a Dunnett’s test of multiple comparisons of the treatments against the control (PBS) was also applied. All formulations induced neutralizing antibody titers, including the commercial vaccine. However, significant differences in neutralizing antibody titers were not detected for DgB + HA (s.c) and the commercial vaccine (*p* ≥ 0.05). DgB + ISA 50 V2 (i.m), (s.c), and the formulation containing aluminum hydroxide (i.m) stimulated significant neutralizing antibody production (*p* < 0.001) when compared with the PBS-vaccinated control. However, no statistical difference was observed among these groups (*p* ≥ 0.05).

In the case of the natural BoHV recombinant strain, the serum neutralization test showed that all formulations, except for DgB + ISA 50 V2 (s.c) and the commercial vaccine, produced significant neutralizing antibody titers (Figure 4) (*p* < 0.001) compared to the control group vaccinated with PBS. A significant neutralizing antibody titer (*p* < 0.05) was also evidenced with the DgB + HA (s.c) and DgB + ISA 50 V2 (i.m) formulations when compared to the group vaccinated with the commercial vaccine. In this case, a statistical analysis of a one-way ANOVA followed by a Dunnett’s test of multiple comparisons of the treatments against the control (PBS) was also applied.

## 4. Discussion

Infectious bovine rhinotracheitis (IBR) and bovine meningoencephalitis are viral infections of great importance for the livestock industry [2]. IBR is characterized by affecting the upper respiratory tract of cattle, causing purulent rhinorrhea, conjunctivitis, loss of appetite, fever, and prostration, while bovine meningoencephalitis has as its main characteristic the death of young animals due to neurological problems [3]. In more advanced moments of infection, the virus can compromise the genital tract of animals, causing balanoposthitis in males and pustular vulvovaginitis in females [1,2,3]. These manifestations are responsible for large economic losses for the livestock industry due to abortions, the weight loss of animals, and reductions in milk production, to which trade restrictions are added [4].

Infections and damage caused by BoHV-1 have been described in reports around the world [24]. In the case of BoHV-5, there are recurrent reports in South America and some sporadic ones in North America and Europe [25]; therefore, it is necessary to improve the success of vaccines to prevent the symptoms produced by both bovine alphaherpesviruses. BoHV-1 exists widely all over the world. Vaccines against BoHV-1 can protect from the clinical signs of BoHV-5 infection due to cross-reactivity [3]. Nevertheless, the World Organization for Animal Health (OIE) reports that although vaccination against BoHV-1/5 can effectively reduce clinical manifestations, the current vaccines do not completely protect against primary infection, latency, and virus reactivation; this is due to the low titer of antibodies that they produce, which is not capable of neutralizing the wild virus. [6]. 

The present study reports the humoral immune response in mice vaccinated against BoHV-1, BoHV-5, and a natural recombinant BoHV-1/BoHV-5 strain using a recombinant chimera of BoHV glycoprotein B as a vaccine antigen. Considering previous studies and the different problems with other prokaryotic organisms [9,10,11] the yeast K. phaffii was selected as a host for the recombinant production of the vaccine candidate “gB Domains”. At shaker scale, the expression of the protein of interest was confirmed, which was efficiently purified through affinity chromatography with a final concentration of 873 µg/mL of the gB domains at the shaker with four cultures of 500 mL [21]. Other studies demonstrated lower purification yields for BoHV glycoproteins using K. phaffii expression of 190 µg/mL or less [9]. The results demonstrate that the candidate vaccine, in combination with different adjuvants, elicits a humoral immune response, being antigenic, immunogenic, and able to induce neutralizing antibodies against BoHV-1, BoHV-5, and the recombinant strain.

The production of neutralizing antibodies was dependent on the inoculation route, with the intramuscular route being the more effective in inducing a humoral response at all time-points evaluated, particularly when the DgB + ISA 50 V2 formulation was administered. Some authors have reported that the time of exposure to the antigen in intramuscular tissues is longer than that obtained by subcutaneous administration, supporting the fact that there are differences in the populations of antigen-presenting cells according to the administration regimen [26,27]. Moreover, the oil-based adjuvant has the characteristic of dosing the antigen in the tissue for a longer time. Thus, the recruitment of antigen-presenting cells is higher [28,29,30], which would possibly explain the obtained results.

Other studies in which a candidate BoHV-1 vaccine composed of envelope glycoprotein subunits was evaluated in different animal models, have shown comparable or lower antibody titers than those obtained in our experiment at day 42 (Figure 2) [10,31,32,33,34,35]. Therefore, our results strongly suggest that our vaccine formulation has the potential to be an effective immunogen for cattle. Furthermore, it is important to note that the inactivated commercial vaccine was unable to induce significant neutralizing antibody titers, which highlights the need for producing more effective, new generation vaccines.

Significant neutralizing antibody titers against BoHV-5 were obtained with DgB + ISA 50 V2 (s.c), DgB + ISA 50 V2 (i.m), and DgB + HA (i.m) formulations (Figure 3). These neutralizing antibody titers obtained against BoHV-5 by these administration routes are consistent with those described by other authors [26,27,28,29,30]. In contrast, other studies reported lower titers of neutralizing antibodies against BoHV-5 [10,11,36]. As previously mentioned for BoHV-1, the commercial vaccine did not elicit significant neutralizing antibody titers against BoHV-5 either.

The higher antibody titers against BoHV-5 compared to BoHV-1 might be explained by the higher scores of possible epitopes recognized by B and T lymphocytes for the PH-domain-like 2 of BoHV-5, as observed when we applied immune-informatics approaches, where scores ranged from 8 to 10, with 10 being the maximum possible score for recognizable B lymphocyte conformational and lineal epitopes [21].

Although the rate of nucleotide substitution in alphaherpesviruses is 3 × 10^−8^ sites per year [37] and the overall rate of protein sequence evolution is 3 × 10^−9^ amino acids per year [38], a report by Maidana et al. (2017) confirms that homologous recombination between BoHV-1 and BoHV-5 can occur under natural field conditions, which could reduce the success of vaccines based on envelope glycoprotein subunits [39]. For this reason, we decided to evaluate the induction of neutralizing antibodies of the DgB vaccine candidate against a BoHV-1/5 natural recombinant strain (Figure 4), demonstrating that all formulations and routes of administration produced significant titers when compared to the unvaccinated control group. Furthermore, DgB + HA (s.c) and DgB + ISA 50 V2 (i.m) induced higher antibody titers than the commercial vaccine. To our knowledge, this is the first report on the evaluation of a vaccine candidate against a recombinant strain of bovine alphaherpesvirus.

Different authors have reported that the possible success of a vaccine against IBR and bovine meningoencephalitis lies in strong humoral immune responses since it has been reported that cellular immune responses are not completely able to prevent propagation, neuroinvasion, and viral reactivation [33,40,41]. In addition, the United States Department of Agriculture considers as immunized a herd that has 80% of animals vaccinated with a neutralizing titer equal to or above 1:8 [42]. Together, our data suggest that the chimeric candidate vaccine (DgB) was safe in vivo and elicited significant neutralizing antibodies against BoHV-1, BoHV-5, and a recombinant BoHV-1/5 strain. These results suggest that the DgB vaccine candidate can be used in the development of vaccines; mainly in those areas where BoHV-1 and -5 are endemic. The next step that is expected for this vaccine candidate will be to test its efficacy in cattle exposed in a challenge experiment to these same strains used here and thus compare it with current vaccines.

## 5. Conclusions

Different studies show the need to search for new vaccine candidates against infectious bovine rhinotracheitis and bovine meningoencephalitis, two diseases caused by bovine alphaherpesvirus 1 and 5, respectively. In this study we propose a bivalent chimeric vaccine candidate composed of B glycoprotein domains from both viral species. Together, our data suggest that the chimeric (DgB) vaccine candidate is safe in vivo, and the results in terms of viral seroneutralization indicate that this vaccine candidate generates a significant humoral immune response compared to the commercial vaccine currently used for prevention. It is hoped that this vaccine candidate can be scaled up in bovines to confirm that its antigenicity is conserved in the target organism of the natural infection of these viruses.

## Figures and Tables

**Figure 1 vaccines-11-01173-f001:**
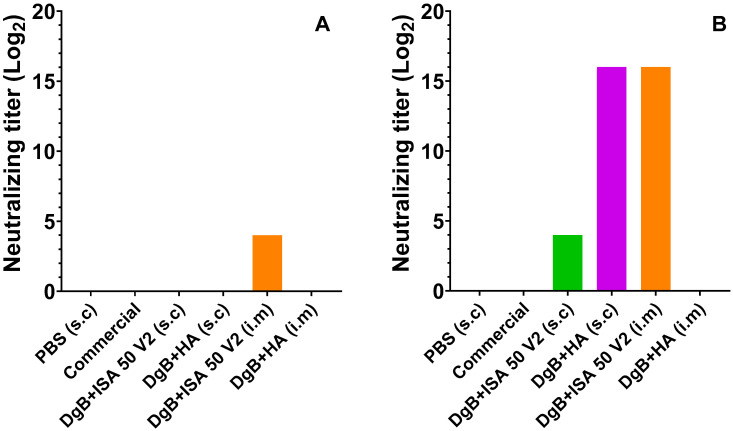
Neutralizing antibody titers against BoHV-1 in mice inoculated with 50 μg DgB formulated with aluminum hydroxide or with oil adjuvant, PBS, or a commercial inactivated vaccine. The titer was determined by viral neutralization assay at different times post-vaccine administration. Antibody titers are represented as log2-transformed data and are expressed as the reciprocal of the highest dilution that completely inhibited virus-induced CPE; (**A**) 14 days post-inoculation, (**B**) 28 days post-inoculation. s.c: subcutaneously; i.m: intramuscularly. PBS: buffer saline solution; Commercial: commercial vaccine; DgB + ISA 50 V2: domain’s glycoprotein B with oily adjuvant; DgB + HA: domain’s glycoprotein B with aluminum hydroxide.

**Figure 2 vaccines-11-01173-f002:**
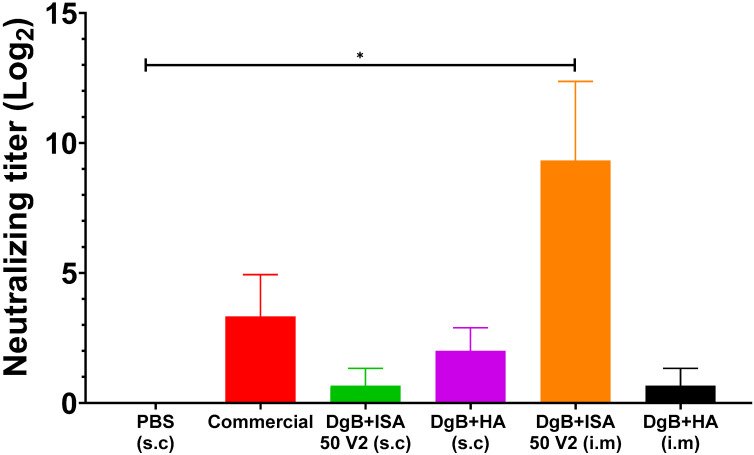
Neutralizing antibody titers against BoHV-1 in mice inoculated with 50 μg DgB formulated with aluminum hydroxide or with oil adjuvant, received PBS, or a commercial inactivated vaccine. Neutralizing antibody titers were determined by viral neutralization assay fourteen days after the third dose administration. The data represent the mean ± S.E.M. of log2-transformed data and are expressed as the reciprocal of the highest dilution that completely inhibited virus-induced CPE. Statistical analysis was performed by one-way ANOVA followed by Dunnett’s multiple comparisons test. * *p* < 0.05. s.c: subcutaneously; i.m: intramuscularly. PBS: buffer saline solution; Commercial: commercial vaccine; DgB + ISA 50 V2: domain’s glycoprotein B with oily adjuvant; DgB + HA: domain’s glycoprotein B with aluminum hydroxide.

**Figure 3 vaccines-11-01173-f003:**
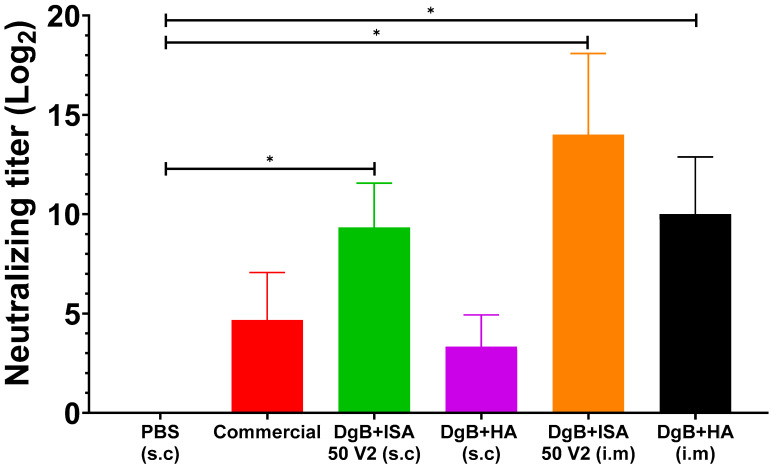
Neutralizing antibody titers against BoHV-5 in mice vaccinated with 50 μg DgB formulated with aluminum hydroxide or with oil adjuvant, or that received PBS or a commercial inactivated vaccine. Neutralizing antibody titers were determined by viral neutralization assay fourteen days after the third dose administration. The data represent the mean ± S.E.M. of log2-transformed data and are expressed as the reciprocal of the highest dilution that completely inhibited virus-induced CPE. Statistical analysis was performed by one-way ANOVA followed by Dunnett’s multiple comparisons test. * *p* < 0.001. s.c: subcutaneously; i.m: intramuscularly. PBS: buffer saline solution; Commercial: commercial vaccine; DgB + ISA 50 V2: domain’s glycoprotein B with oily adjuvant; DgB + HA: domain’s glycoprotein B with aluminum hydroxide.

**Figure 4 vaccines-11-01173-f004:**
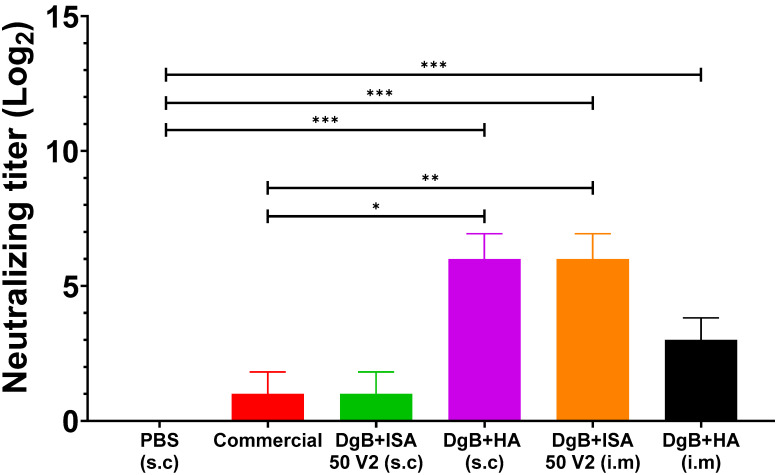
Neutralizing antibody titers against strain recombinant in mice vaccinated with 50 μg DgB formulated with aluminum hydroxide or with oil adjuvant, or that received PBS or a commercial inactivated vaccine. Neutralizing antibody titers were determined by viral neutralization assay fourteen days after the third dose administration. The data represent the mean ± S.E.M. of log2-transformed data and are expressed as the reciprocal of the highest dilution that completely inhibited virus-induced CPE. Statistical analysis was performed by one-way ANOVA followed by Dunnett’s multiple comparisons test. * *p* < 0.05, ** *p* < 0.01, and *** *p* < 0.001. s.c.: subcutaneously; i.m: intramuscularly. PBS: buffer saline solution; Commercial: commercial vaccine; DgB + ISA 50 V2: domain’s glycoprotein B with oily adjuvant; DgB + HA: domain’s glycoprotein B with aluminum hydroxide.

## Data Availability

The data of this study are available upon reasonable request from the authors.

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
