# Peer review of "Humoral Immune Response of Mice against a Vaccine Candidate Composed of a Chimera of gB of Bovine Alphaherpesviruses 1 and 5"

_vaccines, 2023, doi:10.3390/vaccines11071173_

Round 1

Reviewer 1 Report

Barbosa et. al. present the results of a study characterizing the immunogenicity of a bovine herpes virus vaccine.  Recombinant gB was prepared in yeast and tested with a oil based adjuvant and an alum adjuvant.  The outcome measures were neutralizing antibody using a microneutralization test.  A commercially available vaccine was also tested along with controls.  The results in some instances indicate that titers are higher with the experimental vaccines compared to the commercial vaccine.    Overall, the neutralizing titers are not very high so it is not clear from the results that this will be a significant improvement nor is it clear that this vaccine would protect better than current vaccines.  This is not really covered in the discussion and they also don't provide any indication about the efficacy of vaccines that generate the titers they achieved.  Some discussion about the literature on effective neutralizing antibody titers is important.  It is good that they measured neutralizing antibody but there are examples of HSV vaccines where significant protection was generated with antibodies that promote ADCC so the question is why wasn't the antibody characterization more complete?  Other issues include:

The need to show western blots or gels of the recombinant gB so we can assess purity.

In the introduction they mention "capsid wrapping".  If they are talking about envelopment then say that or are they talking about acquisition of the tegument?

The figure legends should include definitions of the groups so the reader doesn't need to refer to the methods

Author Response

Dear Editor,

We thank the editor and reviewers for their insightful comments. We have addressed all comments and feel that the revised manuscript was greatly improved. The main modifications are highlighted in red in the revised manuscript. Our detailed responses to the reviewers can be found as below.

Best regards,

Juan Sebastian Quintero Barbosa

Some discussion about the literature on effective neutralizing antibody titers is important.  It is good that they measured neutralizing antibody but there are examples of HSV vaccines where significant protection was generated with antibodies that promote ADCC so the question is why wasn't the antibody characterization more complete?  Other issues include:

Answer: Taking into account the comment made, we add a reference from the USDA where the titer of protective antibodies considered for a herd is indicated. Line 227-229.

A deeper characterization of antibodies could not be carried out for the serum obtained from the animals, our study had as its main objective the serum neutralization against the 3 viral species, as a plus of our study we have the inclusion of a natural recombinant strain that, to the extent we know it had never been taken into account to test a vaccine candidate.

The need to show western blots or gels of the recombinant gB so we can assess purity.

Answer: This information and images can be consulted in the preliminary article to this, we attach the information.

Quintero Barbosa, J.S., Rojas, H.Y.T., Gonzalez, J. et al. Characterization and expression of domains of Bovine Alphaherpesvirus 1/5 envelope glycoproteins B in Komagataella phaffi. BMC Vet Res 19, 28 (2023). https://doi.org/10.1186/s12917-023-03590-8

In the introduction they mention "capsid wrapping".  If they are talking about envelopment then say that or are they talking about acquisition of the tegument?

Answer: Following the correction, this change was already made in line 57

The figure legends should include definitions of the groups so the reader doesn't need to refer to the methods

Answer: The recommendation was taken into account and the description was added to all image captions

Reviewer 2 Report

The authors described that a chimeric BoHV-1 and BoHV-5 gB expressed in yeast induced neutralizing antibodies against  BoHV-1, BoHV-5, and the recombinant strain in mice. These findings are of interest and provide a promising vaccine candidate for further evaluation in cattle. However, a number of points need clarifying and certain statements require further justification. These are given below.

1. Commercial vaccine are poorly defined. Virus strain, dose, and manufacture should be indicated.

2. Lines 78-80. This sentence should be revised. The yeast was transformed by expression plasmid.

3. Lines 147-149. It is unclear what the authors mean by this sentence.

4. Fig. 1. Panels A and B are not defined.

Minor points:

5. Lines 13-14. This sentence should be revised.

6. Line 16. Onto or to not into.

7. Key words should include bovine alphaherpesvirus 1 and bovine alphaherpesvirus 5.

8. Line 43 and others. Viral life cycle.

9. Line 63. Life stages not viral stages.

10. Line 119. Reference No. should be indicated.

11. Line 123. What are BoHV positive and negative serum?

12. Line 127. USA not MA.

13. Lines 167-169 and others. This statement is too repetitive.

14. Line 271. Than not that.

15. Line 282. Delete bovine.

Author Response

Dear Editor,

We thank the editor and reviewers for their insightful comments. We have addressed all comments and feel that the revised manuscript was greatly improved. The main modifications are highlighted in red in the revised manuscript. Our detailed responses to the reviewers can be found as below.

Best regards,

Juan Sebastian Quintero Barbosa

The authors described that a chimeric BoHV-1 and BoHV-5 gB expressed in yeast induced neutralizing antibodies against BoHV-1, BoHV-5, and the recombinant strain in mice. These findings are of interest and provide a promising vaccine candidate for further evaluation in cattle. However, a number of points need clarifying and certain statements require further justification. These are given below.

1. Commercial vaccines are poorly defined. Virus strain, dose, and manufacture should be indicated.
Answer: This information was added on lines 110-111.

2. Lines 78-80. This sentence should be revised. The yeast was transformed by expression plasmid.
Answer: Sentence revised and changed.

3. Lines 147-149. It is unclear what the authors mean by this sentence.
Answer: In this sentence we wanted to make known the reasons why we only tested the neutralizing antibodies on days 14 and 28 post inoculation, this, due to the low amount of serum obtained by puncture in the cheek of the mice, we opted for test only against BoHV-1 because it has the highest prevalence in Colombia.

4. Fig. 1. Panels A and B are not defined.
Answer. Change made following editor's instructions

Minor points:

5. Lines 13-14. This sentence should be revised.
Answer: Revised sentence.

6. Line 16. Onto or to not into.
Answer: Change made

7. Key words should include bovine alphaherpesvirus 1 and bovine alphaherpesvirus 5.
Answer: Added words

8. Line 43 and others. Viral life cycle.
Answer: Change made

9. Line 63. Life stages not viral stages.
Answer: Change made

10. Line 119. Reference No. should be indicated.
Answer: Change made

11. Line 123. What are BoHV positive and negative serum?
Answer: Negative sera refers to sera tested by ELISA test where the absence of antibodies against BoHV is evidenced and positive sera refers to sera tested by ELISA where Ab against BoHV is evidenced.

12. Line 127. USA not MA.
Answer: Change made

13. Lines 167-169 and others. This statement is too repetitive.
Answer: Change made

14. Line 271. Than not that.
Answer: Change made

15. Line 282. Delete bovine.
Answer: Change made

Round 2

Reviewer 1 Report

Concerns were adequately addressed.